# LARGE LANGUAGE MODEL CRITICS FOR EXECUTION-FREE EVALUATION OF CODE CHANGES

## ABSTRACT

Large language models (LLMs) offer a promising way forward for automating software engineering tasks, such as bug fixes, feature additions, *etc.*, via multi-step LLM-based agentic workflows. However, existing metrics for evaluating such workflows, mainly build status and occasionally log analysis, are too *sparse* and limited in providing the information needed to assess the quality of changes made. In this work, we designed LLM-based critics to derive well-structured and rigorous intermediate/step-level, *execution-free* evaluation proxies for repo-level code changes. Importantly, we assume access to the gold patch for the problem (*i.e.*, reference-aware) to assess both semantics and executability of generated patches. With the gold test patch as a reference, we predict executability of all editing locations with an accuracy of 91.6%, aggregating which, we can predict the build status in 82.1% of the instances in SWE-bench. In particular, such an execution-focused LLM critic outperforms other reference-free and reference-aware LLM critics by 38.9% to 72.5%. Moreover, we demonstrate the usefulness of such a reference-aware framework in comparing patches generated by different agentic workflows. Finally, we open-source the library developed for this project, which allow further usage for either other agentic workflows or other benchmarks.

## 1 INTRODUCTION

Source code within repositories is typically complex and inter-dependent. As a result, code changes (referred to as *edits*) are often spread across multiple functions, classes, and/or files. This makes day-to-day software engineering (SE) tasks – such as fixing bugs or adding new features – onerous for developers. The inter-dependence can also introduce different forms of syntactic, semantic, or logical errors, which may impact multiple locations. Thus, developers often find themselves in iterative editing cycles, where they must repeatedly build, identify and fix failures.

Recent improvements in large language models (LLMs) (OpenAI, 2023; Anthropic, 2024; Touvron et al., 2023) has prompted the use of LLM agents – systems capable of interacting with its environment to make rational decisions – for automating these complex SE tasks through multi-step processes (Zhang et al., 2024; Wang et al., 2024). We refer to such an orchestration of autonomous or semi-autonomous agents as *agentic workflows*. Despite their promise, evaluating the effectiveness and reliability of these workflows poses significant challenges.

Existing metrics for evaluating agentic workflows primarily include build success status and, in some cases, log analyses. However, these are limited. *Firstly*, retrieving such metrics requires setting up a test environment and running the test suite, which is either impossible whenever considering a generic range of code repositories in industrial applications or both laborious and time consuming. *Secondly*, they are too **sparse** and provide a narrow view of the overall performance, which is not sufficient to assess the quality of the changes made. For example, build status does not provide insights into functional correctness or performance under various conditions. *Thirdly*, analyzing logs can be cumbersome and may not provide actionable insights without significant manual interpretation.

These limitations are particularly problematic for partial failures, where the modified code may not even compile, pass unit tests, or the integration test. In such cases, traditional metrics are not sufficiently available to an agent for improving the patch. Therefore, having access to evaluation proxies that assess quality of code changes and are independent of build status after modifications is

**Leaderboard**

| Model | % Resolved | Date | Logs | Trajs | Site |
|---|---|---|---|---|---|
| 🥇 Amazon Q Developer Agent (v20240719–dev) | 19.75 | 2024-07-21 | 🔗 | – | 🔗 |
| 🥈 Factory Code Droid | 19.27 | 2024-06-17 | 🔗 | – | 🔗 |
| 🥉 AutoCodeRover (v20240620) + GPT 4o (2024–05–13) | 18.83 | 2024-06-28 | 🔗 | – | 🔗 |
| 👑 ✅ SWE–agent + Claude 3.5 Sonnet | 18.13 | 2024-06-20 | 🔗 | 🔗 | – |
| 👑 ✅ AppMap Navie + GPT 4o (2024–05–13) | 14.60 | 2024-06-15 | 🔗 | – | 🔗 |
| Amazon Q Developer Agent (v20240430–dev) | 13.82 | 2024-05-09 | 🔗 | – | 🔗 |
| 👑 ✅ SWE–agent + GPT 4 (1106) | 12.47 | 2024-04-02 | 🔗 | 🔗 | 🔗 |
| 👑 ✅ SWE–agent + Claude 3 Opus | 10.51 | 2024-04-02 | 🔗 | 🔗 | – |
| 👑 ✅ RAG + Claude 3 Opus | 3.79 | 2024-04-02 | 🔗 | – | 🔗 |
| 👑 ✅ RAG + Claude 2 | 1.96 | 2023-10-10 | 🔗 | – | – |
| 👑 ✅ RAG + GPT 4 (1106) | 1.31 | 2024-04-02 | 🔗 | – | – |
| 👑 ✅ RAG + SWE–Llama 13B | 0.70 | 2023-10-10 | 🔗 | – | – |
| 👑 ✅ RAG + SWE–Llama 7B | 0.70 | 2023-10-10 | 🔗 | – | – |
| 👑 ✅ RAG + ChatGPT 3.5 | 0.17 | 2023-10-10 | 🔗 | – | – |

– The **% Resolved** metric refers to the percentage of SWE–bench instances (2294 total) that were *resolved* by the model.

Figure 1: A snapshot of SWE-bench leaderboard for different agentic workflows, as of July 2024. (Jimenez et al., 2024) Instance resolution rate provides low discrimination between the different models and is not informative over the failures or partial progress rate.

*desirable*. To this end, we aim to design LLM critics that provide intermediate/step-level information for evaluating agentic workflows – thus establishing a proxy for task progress, useful to compare multiple agentic workflows themselves. Furthermore, by leveraging LLMs in this manner, we can bypass the limitations of traditional metrics that are tied to compilation and execution.

Important, in this work, we assume access to a given solution for the problem (*e.g.* successful commit), which we refer to as a *gold patch* in the remainder of the paper. Among the changes made by a human developer to both source code and tests, we specifically consider the *gold test patch as a reference*, which contains the unit tests useful in determining if an agentic workflow-generated patch resolves an issue. We introduce our *test-centric* framework utilizing *isolated, test-aware*. Importantly, these are always available whenever the problem doesn't require any unit test modifications. We design LLM critics, which leverage a candidate patch against each associated test individually to predict whether the patch helps that test pass or not. Finally, we predict the corresponding build statuses by aggregating the individual assessments across all tests. We relax our assumption by considering reference-free scenario with no ground truth patch access in Section 4.3.

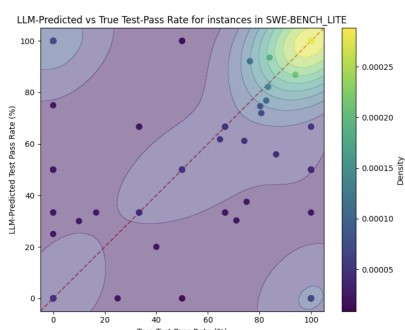

Figure 2: Heatmap graph of predicted and effective test pass rate for all instances of SWE benchmark. Experiment setting and results are further described in 4.1

In essence, our test-centric framework provides a macro-evaluation of the candidate patch as an aggregation of micro-assessments against the individual tests. We demonstrate the effectiveness of our test-aware LLM critics for `factory-code-droid` (Droid, 2024) agent on SWE-bench-Lite (Jimenez et al., 2024) – predicting the individual test oracles and subsequent build outcomes with F1-scores of **91.6%** and **82.1%**, respectively, outperforming the reference-free metrics by 65.5%–75.5% and other reference-aware metrics by 38.9%–68.3% in the latter. In addition, when comparing different agentic workflows, our LLM-predicted task progress proxy rankings perfectly align with the actual ones in **68.8%** of the cases.

Our key findings can be summarized as follows:

(a) We derive micro-evaluation and context enhancement strategies to derive LLM-based optimal evaluation of code patches, enabling fine-grained assessments.

(b) We ground our macro-evaluation approach using micro-evaluation aggregation, leading to better performance than both baseline and other *pure* macro-evaluation strategies.

(c) We open-source the library developed for this project, which allow further usage for either other agentic workflows or other benchmarks.

## 2 AUTOMATIC EVALUATION USING LARGE LANGUAGE MODELS

### 2.1 A PRIMER ON USING LLMS FOR CODE EVALUATION

In the domain of source code, recent work has shown a promise in the use of LLMs for the automated evaluation of code generation tasks, demonstrating their potential to assess code without actual execution (Zhou et al., 2023; Zhuo, 2024; Dong et al., 2023). In this context, two primary classes of such metrics are widely used: *reference-free* and *reference-aware*. The former leverage LLM's understanding of source code and their alignment with the given intent (*i.e.*, natural language description used for code generation) to assess code quality and functional correctness (Zhou et al., 2023; Zhuo, 2024). Conversely, reference-aware metrics compare the generated code with a ground-truth reference implementation, evaluating how closely the former aligns with the latter, thus providing a direct measure for functional correctness. These are effective for the assessment of generated code. However, a task (*e.g.*, GitHub issue) can be resolved in multiple ways by making non-unique code edits at different locations in the repository. Therefore, such automated approaches are not directly extensible for the qualitative assessment of agentic workflow-generated patches.

#### 2.1.1 PROBLEM FORMULATION

Let's assume a set of coding tasks $T \in \mathcal{T}$, typically represented by a goal specified in natural language and a code repository to modify. These tasks can generally represent new feature design, code migration, bug fixing, unit test generation . . . In all generality, we model candidate solutions to the task $T$ as code patches $P = \{p_1, p_2, ..., p_N\}$, representing the before/after difference. Note that the candidate patches $p_i$ can originate from multiple sources, including humans, LLM/agent workflows, or even by artificial perturbations of correct solutions.

The goal of our work is to design an evaluation score $S : p \mapsto \{0, 1\}$ or $S : p \mapsto \mathbb{R}$ that assesses the quality of candidate patches $p_i$ with respect to the task $T$. Importantly, we assume the access to a ground-truth patch $p^\star$ that successfully resolves the task $T$. We allow our score $S$ to depend on that ground-truth patch $p^\star$. In that regard, our problem can be seen as supervised or reference-aware. We relax this assumption in Section 4.3.

In order to measure the quality of our evaluation score with regards to the two goals above, we consider the following metrics:

**Success Discrimination** Granted a set of successful and failed patches, we consider in sections 4.2 and 4.4 classification models for binarized evaluation score $S$ to map the score to a task success criteria such as build status. From this criteria, we assess classical accuracy metrics such as as precision, recall and F1.

**Progress Monotonicity** By using labeled patches whose advancement status is known (e.g., patches generated during agent trajectory) or by artificially perturbing existing patches, we generate set of tuples such $p_1 \preceq p_2$, with respect to a given notion of order and assess that $S(p_1) \leq S(p_2)$ in Section 4.1 and 4.1. Several such proxy notions of order, *i.e.* task advancement, can be designed, either in reference-free or reference-aware mode. For instance, percentage of passing tests, number of line of code changed, number of files modified, correctness of the changes, edit distance between the reference and the candidate, *etc*. We show the limitations of such methods with respect to the initial goal yet illustrate how our method positively correlates with these metrics.

### 2.2 LLM CRITICS FOR EXECUTION-FREE PATCH EVALUATION

Code modifications from an agentic workflow might resolve certain issues while potentially failing to address others, or even introduce new ones. To assess the correctness and effectiveness of such

generated patches, our framework employs *test-centric* LLM critics. These utilize the potentially unseen tests extracted from gold test patch as a reference (*i.e.*, reference-aware) and individually predict whether each of the tests pass or not. For this purpose, we consider two variants of the generated patch:

1. **Context Enhancement**: By default, a patch shows only 3 lines of context around each hunk, which may not be sufficient for accurately predicting test outcomes due to limited understanding of input propagation. To address this, we expand the context to include additional lines that span the entire functions or methods containing the code changes. Such context enhancement provides a more reliable test-centric evaluation of patches.

2. **Source Code**: Next, we extract the functions or methods containing the code changes after applying the patch. We refer to these as *post-commit functions*. The rationale, in this regard, is that the new unseen tests are likely evaluating the post-commit functions.

Such a design represents a *micro-evaluation* of patches, as it assesses the generated patches in the context of the unseen tests, individually. The predictions for each test reveals how the changes affect a particular aspect of the code. This is particularly useful to track progress in potential failures without actual execution, while establishing a comparative framework for different agentic workflows.

To determine the overall build status after applying an agentic workflow-generated patch, we aggregate the individual test oracle predictions. We determine a *build success* if our LLM critic predicts all of the new unseen tests to pass. In contrast, if even one of the tests is predicted to fail, we determine a *build failure*. Note that more ensemble strategies can be explored for this purpose, which is beyond the scope of this work. In summary, our test-centric framework enables an execution-free, *macro-evaluation* of whether the generated patch successfully addresses all intended functionalities by aggregating the micro-evaluations based on new unseen tests.

### 2.2.1 Uncertainty Quantification with Black-Box Confidence Measures

As described in Section 2.2, we first formulate our micro-evaluation of patches using LLM critics as a binary classification task. To calibrate these predictions, we assume that all parameters during inference are unknown. We elicit confidence estimates by prompting the LLM to express its confidence in unseen test pass/fail prediction as a value between 0 and 100. When combined with Chain-of-Thought (COT) prompting (Wei et al., 2022), such *verbalized* confidence measures have been shown to be useful for improving the reliability of the LLM's predictions (Xiong et al., 2024).

## 3 Experiment Setup

### 3.1 Dataset

SWE-bench (Jimenez et al., 2024) is a benchmark comprising real-world software engineering tasks, with each instance containing pairs of GitHub issues and corresponding pull requests. While the former describes the desired changes to the codebase, the latter includes the *actual* code changes made by human developers in resolving the issue and the test cases to validate the changes. Here, the goal of the agentic workflow is to interact with the unfixed repository snapshot, and attempt to fix the issue. The generated patch is tested against the new unseen tests as well as existing tests impacted. In this paper, we consider a canonical subset of SWE-bench (dubbed SWE-bench-Lite) containing 300 instances collected from 11 popular Python projects, where the gold change patch contains at most 3 edits in a single file.

### 3.2 Models and Agentic Patches

We conduct our main experiments using `claude-3-opus` as the LLM critic. As patches, we utilize the generated patches from `factory-code-droid` (Droid, 2024), `sweagent-gpt4` (Jimenez et al., 2024), `Gru` (Gru.ai, 2024) and `codestory-aide-mixed` (Aide.dev, 2024). These are selected to be representative agentic trajectories over the benchmark, as further discussed in Section 4.4. However, note that our evaluation framework is both agentic workflow and LLM-agnostic. In Section 4.4, we compare our test-centric framework for patches from multiple agentic workflows, and in Section 4.5, we compare with different LLMs. We will open-source the library developed for

Table 1: Performance comparison of our test-centric framework in test oracle prediction

| Approach | Candidate Patch | Evaluation Metrics (in %) | | | |
|---|---|---|---|---|---|
| | | *Accuracy* | *Precision* | *Recall* | *F1-Score* |
| Random | – | 75.0 | 84.7 | 85.9 | 85.3 |
| Isolated, Test-Aware | *post-commit functions* | 69.3 | 83.6 | 79.1 | 81.3 |
| | **± *function-level*** | **84.8** | **85.4** | **98.8** | **91.6** |

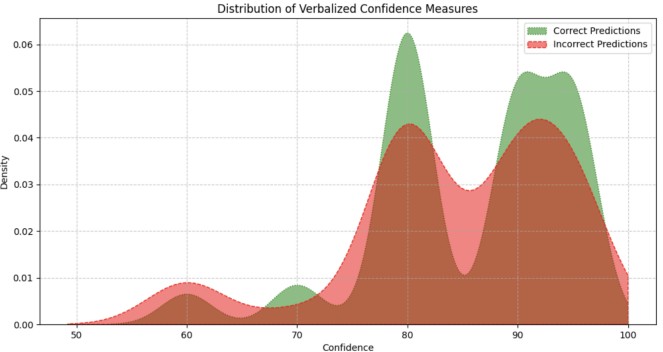 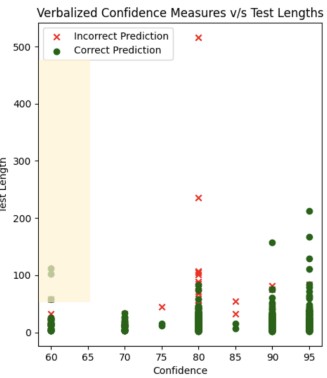

Figure 3: (*left*) Distribution of verbalized confidence scores for both correct and incorrect isolated, test-aware LLM critic predictions. (*right*) Plot of these confidence scores against test character length, as a proxy to test complexity. Here, the highlighted region indicates the confidence threshold.

this project, which allow further usage for either other agentic workflows or other benchmarks as well as full reproducibility of our results.

## 4 EVALUATING THE EVALUATORS

### 4.1 LLM CRITICS FOR MICRO-EVALUATION OF PATCHES

As described in Section 3.1, a task instance contains gold tests that an agentic patch is expected to pass. To enable a micro-evaluation of patches (*i.e.*, in predicting test oracles), we leverage isolated, test-aware LLM critics that are given access to each of the unseen tests independently as a reference.

In this experiment, we evaluate our test-centric framework against a random baseline. Except when using a less effective agentic workflow (*e.g.* `RAG + ChatGPT-3.5` on the SWE-bench leaderboard), where a higher number of tests fail, other agents have significantly more passing ones (ratio of passing to failing tests for `factory-code-droid` is 85:15). This indicates errors are typically concentrated in fewer editing locations within a multi-hunk patch. We account for such a class imbalance by weighing the probabilities for our baseline proportional to the class frequencies.

In Table 1, we report the performance of our isolated, test-aware LLM critics using both patch variants for test oracle prediction. We can see that the LLM critic with context-enhanced patches performs the best, outperforming other baselines by 7.4% to 12.7%. Interestingly, the imbalance in passing and failing tests yields a strong random baseline, which records better performance than even the LLM critic with source code (*i.e.*, post-commit functions). However, as we show in Section 4.2, this is not useful in predicting build outcomes. Upon further analysis, we observed that the LLM critic utilizing source code instead of context-enhanced patches helps identify failing tests better. In contrast, the latter helps identify around 25% more passing tests. This may be because, in this case, the LLM understands the purpose of the test in the context of the changes better.

**Confidence Scores v/s Test Complexity.** Here, we explore the verbalized confidence scores from the isolated, test-aware LLM critic, and its relationship with test complexity. In Figure 3 (*left*), we see a clear pattern: the LLM critic tends to make *more* correct predictions with a high confidence than incorrect ones, thus suggesting that the LLM's verbalized confidence scores are a reliable in-

dicator of its predictions. Using test character lengths as a proxy to complexity, in Figure 3 (*right*), we can see that this is particularly true for tests with lower complexity. Based on these analyses, we chose to automatically assume *failures* when the LLM critic assesses a patch with **low confidence** ($\leq 65\%$) for tests with **high complexity** (test length $> 50$). We observed that such thresholding helps improve the specificity (*i.e.* true negative rate) by 24.1% (from $8.3\%$ to $10.3\%$). We differ to further work an extended discussion on the usage of the test complexity and model confidence to improve the prediction quality.

### 4.2 LLM Critics for Macro-Evaluation of Patches

To enable the macro-evaluation of patches (*i.e.*, in predicting build outcomes), in our test-centric framework, we next aggregate the predictions from the isolated, test-aware LLM critics (as in Section 4.1). If even one test among all unseen tests is expected to fail, we predict a build failure. Otherwise, we predict that the patch would successfully pass the build.

#### 4.2.1 Baselines

We select multiple baselines to evaluate our test-centric framework in build status prediction:

(a) *Random*: First, to establish a reference for the more sophisticated LLM critics, we aggregated the predictions from the random baseline in Section 4.1, with the assumption that even a single failing test prediction results in a build failure. Such an approach helps us measure the importance of aggregating reasoning-based test oracle predictions from LLM critics as opposed to random ones, in determining build outcomes.

Every task instance in SWE-Bench comes with a gold patch, containing: the actual code changes human developers made to resolve the issue, and the test cases that validate the changes. In our test-centric framework, we consider only the test cases as the reference. Here, aiming to evaluate whether the generated patch and the *gold change patch* are equivalent, we use the latter as the reference.

(b) *Edit Distance*: In this baseline, we leverage a pre-trained code language model, Code-BERT (Feng et al., 2020), to retrieve the embeddings for both the generated patch and the reference. Next, we compute the cosine similarity between the two to quantify the similarities between their semantic contents. To determine the build outcome, we perform a grid search on the validation set to identify an optimal threshold, which we then apply to all task instances.

(c) *Change-Aware*: Here, we design LLM critics probing for patch equivalence. Given the candidate and gold change patch pairs, these determine whether they would result in the same functional outcome. To this end, we assume two patch variants: one, *default* patches with 3 lines of context around each hunk; two, $\pm$ function-level patches (as described in Section 2.2).

Next, to assess the importance of aggregating micro-evaluations, we compared with:

(d) *Holistic, Test-Aware*: In this baseline, we design LLM critics to predict the collective outcome of all new unseen tests. A positive prediction indicates that all tests pass, *i.e.*, build success. Conversely, a negative prediction indicates at least one of the tests fails, signifying a build failure. To this end, the holistic, test-aware LLM critics take the generated patch and all corresponding reference tests as inputs. Here, we consider both patch variants (as in Section 2.2).

#### 4.2.2 Experimental Results

Table 2 shows the performance comparison for build status prediction. We can see that aggregating the test oracle predictions from isolated, test-aware LLM critics yields the best performance, predicting the build outcomes with an F1-score of 82.1%. Notably, we observe an improvement over all baselines by 38.9% to 159%, and over other LLM critics by 38.9% to 68.2%.

We can see that aggregating the random baseline in Section 4.1 performs poorly. This asserts the complexity of the task, and explains the need for aggregating reasoning-based test oracle predictors. Furthermore, we improve upon the edit distance-based approach by 72.1%. Interestingly, we observed that this baseline did not capture even a single build failure. This could be due to Code-BERT's lack of understanding of the structure of patches, highlighting the limitations of directly applying pre-trained code language models to code changes.

Table 2: Performance comparison of our test-centric framework in build status prediction compared to other reference-aware baselines.

| Approach | Candidate Patch | Evaluation Metrics (in %) | | | |
|---|---|---|---|---|---|
| | | *Accuracy* | *Precision* | *Recall* | *F1-Score* |
| Random | – | 62.7 | 71.1 | 76.9 | 31.7 |
| Edit distance | *default* | 31.3 | 31.3 | 100 | 47.7 |
| Change-Aware | *default* | 65.7 | 47.1 | 76.6 | 58.3 |
| | $\pm$ *function-level* | 65.0 | 46.6 | 80.9 | 59.1 |
| Holistic, Test-Aware | *post-commit functions* | 44.3 | 34.5 | 86.2 | 49.2 |
| | $\pm$ *function-level* | 35.0 | 32.4 | 98.9 | 48.8 |
| Isolated, Test-Aware | *post-commit functions* | 64.7 | 73.1 | 76.9 | 74.9 |
| | $\pm$ **function-level** | **71.4** | **72.1** | **95.4** | **82.1** |

Importantly, we see that using *gold change patch* as a reference (*i.e.*, change-aware) instead of the *gold test patch* leads to a decrease in performance by 40.8%. This is possibly because a problem can be solved by the LLM and a human developer very differently, resulting in varied changes across different editing locations. As a result, comparing with the gold change patch might not always be helpful. Furthermore, many such plausible implementations also prohibits aggregating the micro-assessments of change-aware approaches, as it might not be possible to establish a one-to-one correspondence between the editing locations in the candidate and reference patches.

When compared against the holistic test-aware LLM critics, we see that aggregating the isolated test-aware LLM critics improves performance in F1-score by 52.2% and 68.2%, respectively, with source code and context-enhanced patches. Notably, in both experiment settings, using patches instead of source code helps correctly predict more build successes. Finally, comparing the macro-evaluations in change-aware and holistic test-aware baselines, we can see that using gold change patch as a reference is more helpful than gold test patch. However, this could possibly be due to the higher number of complex tests in the test-centric approaches (see Figure 3, *right*), which degrades the LLM's determination.

In summary, based on our observations on the change-aware and holistic test-aware LLM critic baselines, we can draw the following conclusions: (a) aggregating fine-grained assessments of candidate patches leads to better performance than evaluating them as a whole, (b) using tests as a reference proves more effective than code changes, more so because these enable fine-grained assessments.

### 4.3 Reference Helps, but is Not Always Available!

In our test-centric framework, we use the new unseen tests as the reference. However, in real-world scenarios, this assumption might not hold. In this experiment, aiming to assess the importance of such a reference, we compare against two reference-free approaches.

All SWE-bench dataset instances contain: (a) *Problem Statement*, which represents a natural language description of desired changes to the codebase, and (b) *Hints*, which represent natural language suggestions on how to solve the problem. These are often used by agentic workflows in generating candidate patches. Here, we design baselines that use the *Problem Statement*, and *Problem Statement + Hints*, to determine if the generated candidate patch helps solve the task description.

In Table 3, we report the results for the reference-free baselines. We can see that our test-centric framework outperforms both baselines by 72.5% and 65.5%, respectively. Note that hints usually contain low-level details, such as pseudo-code suggestions to the original human task worker. While this might be useful to generate candidate patches with agents, these are not particularly useful to evaluate the generated patches. This is also reflected in the comparison of both baselines, with negligible improvements upon including the hints to the LLM critic. Moreover, we can see that enhancing the patches with additional context does not help either, showing inconsistent trends.

Based on these results, we can conclude that reference-free metrics are not very useful to evaluate agentic patches. To this end, we posit that to extend the agent application boundaries, it is important

Table 3: Comparison of Reference-free evaluation with various input and context levels. Although they're not a reference-free method, test-centric results are provided to highlight the gap between the two approches.

| LLM Inputs | Candidate Patch | Evaluation Metrics (in %) | | | |
|---|---|---|---|---|---|
| | | *Accuracy* | *Precision* | *Recall* | *F1-Score* |
| Problem Statement | *default* | 35.3 | 31.9 | 93.6 | **47.6 (↓ 72.5%)** |
| | *± function-level* | 35.3 | 31.6 | 91.5 | 47.0 |
| Problem Statement + Hints | *default* | 44.7 | 34.4 | 84.0 | 48.8 |
| | *± function-level* | 43.4 | 34.4 | 88.3 | **49.6 (↓ 65.5%)** |
| Test-centric (*ours*) | *± function-level* | 71.4 | 72.1 | 95.4 | **82.1** |

Table 4: Test-aware evaluation comparison for patches from different agentic workflows on SWE Benchmark

| Task | Agentic Workflow | Evaluation Metrics (in %) | | | |
|---|---|---|---|---|---|
| | | *Accuracy* | *Precision* | *Recall* | *F1-Score* |
| Micro-evaluation | Codestory Aide Mixed | 87.1 | 87.4 | 99.3 | 93.0 |
| | Factory Code Droid | 85.2 | 85.8 | 98.9 | 91.9 |
| | SWE-Agent + GPT-4 | 81.4 | 83.8 | 95.9 | 89.4 |
| | Gru | 86.1 | 87.0 | 98.46 | 92.4 |
| Macro-evaluation | Codestory Aide Mixed | 76.3 | 77.3 | 96.4 | 85.8 |
| | Factory Code Droid | 72.0 | 72.6 | 95.3 | 82.4 |
| | SWE-Agent + GPT-4 | 66.1 | 65.2 | 91.8 | 76.2 |
| | Gru | 73.0 | 74.5 | 93.5 | 82.9 |

for the LLMs to *learn to improve themselves* via nuanced and generic evaluation proxies beyond build statuses and log analyses. This potentially sidesteps the existing need for agent designers to manually modify parts of the workflow. By leveraging benchmarks with "trusted" metrics (*e.g.*, SWE-bench), we argue that our test-aware LLM critics act as a first step in this direction, helping capture the macro-impact of micro-changes useful to improve agents autonomously.

## 4.4 COMPARING AGENTIC PATCHES WITH TEST-AWARE LLM CRITICS

As noted earlier, in the intrinsic evaluation of our test-centric framework (Sections 4.1– 4.3), we assess the agentic workflow-generated patches from `factory-code-droid` (Droid, 2024). In this experiment, we extend our evaluation to patches from other agentic workflows from the SWE-bench-Lite leaderboard, namely, `gru` (Gru.ai, 2024), `codestory-aide-mixed` (Aide.dev, 2024), and `swe-agent+gpt4` (Jimenez et al., 2024). While the former resolves 31% of all task instances of the Lite SWE-bench, the latter resolve 36%, 43% and 18%, respectively, as of September $30^{th}$, 2024.

In Table 4, we report the results in predicting test oracles (*i.e.*, micro-evaluation) and build status outcomes (*i.e.*, macro-evaluation) for all three agentic workflows. A notable trend is that the LLM critics demonstrate slightly better performance when evaluating workflows with higher-quality patches. This suggests that the LLM critics are sensitive to agent quality, likely because workflows producing better patches lead to higher test pass rates and more consistent results. This is further illustrated in Figure 4 (*left*), where we plot the LLM-predicted test pass rates based on the test oracles predicted by our test-aware LLM critics for all three agentic patches. We can see that patches from `swe-agent+gpt4`, which are of lower quality and fail more tests, correspond to lower test pass rates, particularly evident in the range of $0.0 - 0.6$. On the other hand, patches from `codestory-aide-mixed`, which are of relatively higher quality, achieve higher test pass rates, with greater counts between 0.9 and 1. These trends are as expected, highlighting the reliability of using predicted test pass rates as a proxy for evaluating progress across different workflows.

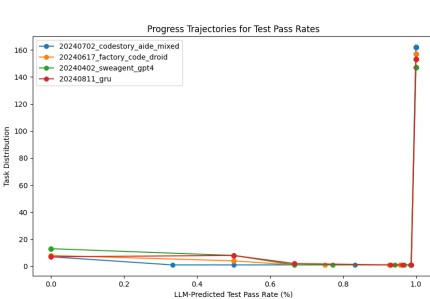 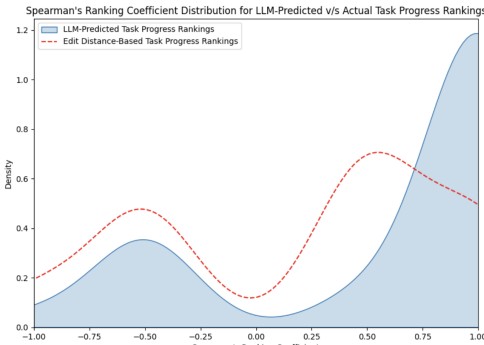

Figure 4: (*left*) Task instance distribution by LLM-predicted test pass rates. (*right*) Density plot of the Spearman's ranking coefficient between LLM-predicted and actual task progress rankings, for each test instance.

Table 5: Performance comparison of our test-centric framework with different LLMs

| Task | Model | Evaluation Metrics (in %) | | | |
|------|-------|----------|-----------|--------|----------|
| | | *Accuracy* | *Precision* | *Recall* | *F1-Score* |
| Micro-evaluation | `claude-3-sonnet` | 75.7 | 85.2 | 86.1 | 85.7 |
| | `claude-3-opus` | **84.6** | 85.1 | **98.9** | **91.5** |
| | `claude-3-5-sonnet` | 84.5 | **85.2** | 98.7 | 91.4 |
| Macro-evaluation | `claude-3-sonnet` | 66.7 | 74.1 | 79.2 | 76.5 |
| | `claude-3-opus` | **70.6** | **71.3** | 96.0 | **81.8** |
| | `claude-3-5-sonnet` | 70.2 | 71.1 | **95.4** | 81.5 |

One of the goal is to use for a given instance labeled patches $p_1, \ldots, p_N$ whose advancement status is known: for instance, test pass rate provides an order over the patches such that $p_1 \preceq \cdots \preceq p_N$. Here, we aim to assess how our evaluation score $S$ correlates with this order.

Accordingly, we ranked the agentic workflows in decreasing order of LLM-predicted test pass rates for all corresponding candidate patches. We then compared these rankings with the true rank-orders of the workflows and computed the Spearman rank-order correlation ($\rho$) between them, to quantify the degree of agreement between the predicted and true rank orders. In Figure 4 (*right*), we show the distribution of these correlation coefficients for all task instances. As a baseline, we also calculated the edit distance between the candidate and true patches and ranked them based on cosine similarity, which is highlighted in red. For 68.1% of the task instances, the predicted rank order perfectly aligned with the true rank order (*i.e.*, $\rho = 1$). Notably, the baseline rank order proves to be less reliable, as evidenced by consistently higher counts when $\rho < 0$. Therefore, the rank orders based on LLM-predicted test pass rates generally align well with the true orders, making them a robust proxy for evaluating and comparing the agentic workflows.

## 4.5 HARNESSING DIFFERENT LLMS IN TEST-AWARE EVALUATION

In Table 5, we compare the performance of our test-aware LLM critics in both micro and macro-evaluation settings for three LLMs from Anthropic (Anthropic, 2024): `claude-3-sonnet`, `claude-3-opus`, `claude-3-5-sonnet`. We can see that both `claude-3-opus` and `claude-3-5-sonnet` achieve comparable performance in predicting test oracles, which is 6.8% better than when using `claude-3-sonnet`. By aggregating these micro-assessments, with `claude-3-opus`, our test-aware LLM critic predict build outcomes the best, with an F1-score of 81.8%. These findings underscore the LLM-agnostic nature of our test-aware LLM critics, although we differ to further work an extensive comparison of different LLMs for this task.

## 5 RELATED WORK

**Large Language Models for Code.** Recent software engineering (SE) research has focused on the use of machine learning-based approaches for SE tasks including program synthesis (Li et al., 2022; Nijkamp et al., 2023), vulnerability detection (Fu & Tantithamthavorn, 2022), automated program repair (Li et al., 2020; Ahmed & Devanbu, 2023), test generation (Schäfer et al., 2023), *etc*. These have traditionally been limited to code snippets (often at the method-level) extracted from software repositories. With the advances in large language models (LLMs), there has been a shift in focus towards extending these to the repository-level, for SE tasks like code (Bi et al., 2024; Deng et al., 2024; Pan et al., 2024) and patch (*i.e.*, code change) generation (Zhang et al., 2024; Bairi et al., 2024). In this work, we design test-aware LLM critics to evaluate code changes.

**Benchmarks for Repository-Level Coding Tasks.** Building on method-level (Chen et al., 2021; Austin et al., 2021) and class-level (Du et al., 2023) code generation benchmarks, there has been a rise in repository-level code generation benchmarks in academia. CrossCodeEval (Ding et al., 2023), CoderEval (Yu et al., 2024), RepoBench (Liu et al., 2024) support multilingual code generation tasks utilizing cross-file context extracted from real-world open-source repositories. Extending beyond code completion, SWE-bench (Jimenez et al., 2024) introduces a broader set of challenges involving patch generation, grounded in real-world software engineering tasks like bug fixing, feature addition or enhancement, *etc*. However, SWE-bench is limited to Python task instances. As a first step toward multilingual support, SWE-bench-java (Zan et al., 2024) was developed to extend this framework to Java. We evaluate our LLM critics on the Python task instances in SWE-bench. However, these can be easily extended to new programming languages and repositories, providing a foundation for assessing the code execution-specific understanding of LLMs across other languages.

**Automated Evaluation of Large Language Models in Coding Tasks.**

Traditionally, the generated code can be evaluated statically, in terms of software quality (*e.g.* readability, complexity (Oman & Hagemeister, 1992)). In particular, code changes in a repository can be assessed via program differencing (Apiwattanapong et al., 2004) and change impact analysis (Ren et al., 2005) – useful to determine the effect of a change on the rest of the repository. However, none of these approaches account for the correctness of the generated code or patches.

By matching against a reference solution, semantic-based metrics such as BLEU (Papineni et al., 2002), ROUGE (Lin, 2004), and CodeBLEU (Ren et al., 2020) or neural based metrics such as CodeBERT (Feng et al., 2020) help establish match-based evaluation proxies. However, these are limited to source code and do not capture program semantics well nor correlate efficiently with human judgment (Eghbali & Pradel, 2004; Tran et al., 2019). Recent work has proposed the use of LLMs for such an evaluation, thus helping establish execution-free evaluation proxies which probe for correctness. These include both reference-free (Zhou et al., 2023; Zhuo, 2024) and reference-aware approaches (*i.e.*, those utilizing human developer-written code or tests) (Dong et al., 2023). However, these are not extensible for the evaluation of patches and still exhibit limited efficiency. In this work, we design both reference-free and test-reference-aware evaluation proxies (the latter significantly outperforming the former).

## 6 CONCLUSION

As a conclusion, we designed LLM-based critics to derive *execution-free* evaluation proxies for repo-level code changes. With the gold test patch as a reference, we predict executability of all editing locations with an accuracy of 91.5%, aggregating which, we can predict the build status in 81.8% of the instances in SWE-bench, outperforming other reference-free and reference-aware LLM critics by 38.9% to 72.5%. Most notably, we observe that aggregating fine-grained assessments of candidate patches leads to better performance than evaluating them as a whole and that using tests as a reference proves more effective than code changes, more so because these enable fine-grained assessments.

Natural extensions of our work include investigating benchmarks from other programming languages, directly incorporating this execution score in agentic framework by relaxing the reference-aware character of our evaluation, and utilizing the LLM input to proactively design better tests.

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
