# OpenReview forum: "Large Language Model Critics for Execution-Free Evaluation of Code Changes"
_ICLR.cc/2025/Conference — Submitted to ICLR 2025_

### Official Review · Reviewer_EhhE · 2024-11-03

**Soundness:** 3
**Presentation:** 3
**Contribution:** 3
**Rating:** 5
**Confidence:** 3

**Summary:**

This paper introduces a novel approach for evaluating code patches made by Large Language Model (LLM) agents using LLM-based critics. The key innovation is developing execution-free evaluation proxies that can assess repository-level code patches without requiring actual compilation or test execution. The authors propose a test-centric framework that leverages gold test patches as references to predict both individual test outcomes and overall build status. Their method achieves 91.6% accuracy in predicting test executability and 82.1% accuracy in predicting build status on the SWE-bench dataset, significantly outperforming reference-free and other reference-aware approaches.

**Strengths:**

- The studied problem of evaluating code changes without execution is interesting.
- The paper is well-written and easy to follow.
- The proposed method is effective.
- Thorough ablation studies and analysis of different components.

**Weaknesses:**

There are several major weaknesses in this paper:

1. The requirement for gold patches in the proposed method limits its applicability in practice. Although it also discusses a reference-free situation, the accuracies below 0.5 are far from practical use.
2. The proposed method is only compared to a simple Edit Distance baseline in the experiments. If given the gold patches, various metrics like CodeBLEU and ROUGE could also provide a way to evaluate the similarity between the deleted and inserted code snippets in the generated output and gold patches. However, such baselines are missing from the experiments. Including these comparisons in the experiments and discussing the results would provide a more comprehensive evaluation of the proposed method's advantages.
3. To evaluate the proposed method across different LLMs, this paper uses three variants of Claude models. However, the choice of using models from the same series may limit the generalizability of the proposed method. It would be beneficial to evaluate the method on more diverse LLMs, such as GPT-4 or other open-source models, to demonstrate its effectiveness.
4. In what scenarios would the proposed method fail? There appears to be a lack of case studies or discussions on the success and failure cases of the proposed method. It would be helpful to provide examples that illustrate the benefits and limitations of the proposed method compared to other baselines. Additionally, I look forward to seeing example inputs and outputs of the LLM evaluator for code patches to better understand the proposed method.

And some minor issues:

1. The organization of Section 2 is somewhat confusing and lacks a figure or algorithm to illustrate the proposed method, making it difficult for readers to understand. There is only one subsubsection under each subsection, which obscures the structure of the section. I recommend that the authors consider adding a figure or algorithm table to help readers better grasp the proposed method and reorganize the section for improved clarity.
2. The authors claim to open-source the code in Lines 25 and 111 as an important contribution. However, the code is not provided in the supplementary material or the paper with anonymous links, making it difficult for me to evaluate the artifacts.
3. As claimed in Line 192, the proposed method prompts LLMs to generate evaluations of code patches. However, the details of how the LLMs are prompted are not clear. It would be helpful to provide the exact prompt in the Appendix.
4. The left figure in Figure 4 is too small, and there are timestamps in the caption that seem unnecessary. It is also recommended to use a larger font size for the text in Figures 3 and 4.

**Questions:**

Please address the concerns in the "Weaknesses" section.

---

### Official Review · Reviewer_3RwT · 2024-11-03

**Soundness:** 2
**Presentation:** 2
**Contribution:** 3
**Rating:** 3
**Confidence:** 4

**Summary:**

The paper studies the problem of performing execution-free evaluation of repository-level code changes made by programming agents. Assuming access to ground truth code changes, the paper uses LLM critics (akin to generative verifiers) to assess the correctness of the proposed code changes without actually executing the code. It adopts a "test-centric" granular framework comprising micro-evaluation per test case aggregated to provide a macro "problem-level" evaluation.

**Strengths:**

* Execution-free evaluation for code generation is an important and under-explored problem. Execution-based evaluation is often a challenging engineering problem, and costly. Using gold-patch-guided neural verifiers addresses attempts to address this. Notably, the problem (and proposed approach) has applications beyond benchmarking; it can be used to collect inexpensive 'reward-model' feedback for arbitrary GitHub repositories.
* Intuitive and novel approach to using LLM workflow for a test-driven evaluation of code changes. The micro-evaluation aggregation-based method is more sensible, provided that the evaluations are well-calibrated.

**Weaknesses:**

* **Accuracy of LLM critics.** The paper evaluates gold-patch-guided LLM critics aggregated over test cases; however, LLM-based judges and verifiers are usually quite inaccurate and miscalibrated, even for simple programming problems like those in HumanEval or LeetCode. For example, [1] reported about 50% accuracy for open-source models serving as LLM critics, while GPT-4 achieves only 70-80% accuracy. This raises doubts about the feasibility of execution-free approaches to more complex software code changes studied in this paper.

* **Baseline experiment on standard programming evaluations.** Building on the previous point, standard programming benchmarks like HumanEval or programming contest problems can be formulated as 'code change' problems—for example, given a function with a docstring, insert the necessary code. Understanding the effectiveness of the approach on such "simpler" settings might provide a more grounded understanding of the strength of the approach.

* **Overfitting to SWEBench.** The authors use SWE-Bench-Lite as the sole evaluation benchmark; however, as they acknowledge, SWE-Bench-Lite is imbalanced, with the majority of tests passing. This raises concerns about the generalization of the approach. Specifically, in the micro-evaluations, authors observe a 98% recall -- potentially due to bias in LLMs to respond correct [1]. This aligns with the evaluation suite with high positives potentially inflating the results. It is unclear if this approach will generalize to more challenging benchmarks where a smaller fraction of tests pass and should be evaluated (say on SWEBench full suite).

* **Calibration on the test benchmark.** Authors recalibrated model confidences -- capping a 65% confidence on model responding YES. However, this number uses private knowledge about the correctness of the patches during micro-evaluation recalibration on the SWEBench test set. This raises concerns about the potential for data leakage and the validity of the evaluation results.

* **Pass to Fail tests.** In many cases, a programming agent solution can fail if it introduces a bug in an already passing testcase. It seems this is not handled since the approach only handled "newly introduced" tests in the PR

* **Access to clean pull requests (PRs) is assumed.** SWE-Bench, problem instances (PRs) are cleaned via execution to map the tests and code changes using a Fail-to-Pass strategy. From my understanding, the authors assume access to this information for setting up LLM critics. While this approach works for benchmarks, real-world PRs can be messy, containing unrelated changes to tests and code. This complexity may require at least one execution round to collect the necessary information, limiting the real-world applicability of the work focusing on execution-free nature.

* **Undefined or inconsistent Terminology.**
  * 'build status' is never defined and it is unclear if authors mean simply "all tests passing" or something beyond that
  * In Table 1, settings in the candidate patch column are not properly explained -- `+- function-level` supposed to mean context-enhanced patches is not clear from the table.
  * In Table 2, change-aware vs test-aware is not immediately clear from the description and could be explained with examples.
  * 'code changes,' 'patches,' and 'edits' are used interchangeably without clarification and should be normalized to one aspect


References.

1. The Counterfeit Conundrum: Can Code Language Models Grasp the Nuances of Their Incorrect Generations?

**Questions:**

* I would recommend evaluating the approach over more benchmarks ranging from simple coding problems, competition problems, and harder software engineering problems (SWEBench full set) to reinforce the results.

Minor:

* The paper mentions that "build status does not provide insights into functional correctness or performance under various conditions," which is confusing since build status often involves passing the test suite, which assesses functional correctness. Could you clarify what you mean by "build status" and how it relates to functional correctness in your context? Additionally, since the approach is compared against test success rate on SWEBench, it is not clear how the proposed approach performed better than functional correctness

* The authors mention "modified code may not even compile, or pass unit tests or the integration test. In such cases, traditional metrics are not sufficiently available to an agent to improve the patch". However, the agent cannot access the ground truth test cases and should not use the signals from evaluation proxies to improve the patch. Therefore, I do not follow this argument.

* Typos:
  * Line 80: Important -> Importantly
  * Line 274: differ -> defer
  * Some sentences in the paper can be restructured to flow better.

---

### Official Review · Reviewer_JASr · 2024-11-04

**Soundness:** 3
**Presentation:** 2
**Contribution:** 1
**Rating:** 3
**Confidence:** 5

**Summary:**

This paper proposes a method for evaluating code changes made by agentic workflows in software engineering. The authors introduce LLM-based critics that utilize a test-centric, execution-free framework for assessing candidate patches, using a "gold patch" as a reference to predict the build status. The framework is evaluated on SWE-bench, demonstrating acceptable performance over baseline and alternative evaluation methods in predicting code executability and build outcomes.

However, this paper lacks a clear algorithm/overview flow chart to show their methodological contributions - or, in other words, there is no eye-catching innovation except for the relative rigor of the indicators and experiments used in the comparison between LLM evaluation and human evaluation. This is more like an Empirical Study to reveal that LLM critic can do a good job in patch code generation. But if that's all, exploring the code volume and domain scope of the patch should be fully discussed. The granular discussion based only on SWE-bench is far from sufficient.

**Strengths:**

1. **Evaluation Framework**: The introduction of LLM-based critics for execution-free assessment of code patches trys to fill a gap in software engineering evaluation, as traditional methods like build status or log analysis require execution environments. This framework's approach to utilizing a "test-centric" structure is commendable, as it allows a fine-grained evaluation of patch quality at a functional level, aligning with real-world scenarios where compiling and testing environments may be costly or unavailable.

2. **Comparative Rigor**: The authors offer an extensive experimental setup, comparing the LLM critic-based framework against multiple baselines. Their approach includes micro-evaluations for individual test predictions and macro-evaluation aggregations for build predictions, which demonstrates the framework's flexibility and adaptability across multiple agentic workflows and LLM models.

3. **Performance on SWE-bench**: The proposed framework outperforms reference-free and some reference-aware baselines on SWE-bench, achieving a prediction accuracy of 91.5% for executability and 82.1% for build status. The improvements in prediction accuracy, particularly compared to edit-distance or change-aware baselines, are notable and validate the effectiveness of the LLM.

**Weaknesses:**

1. **Technical Description**: The technical pipeline of the test-centric framework is intricate, and certain steps could be explained with more clarity and visualization. For example, the mechanisms by which LLM critics predict test pass/fail outcomes based on context enhancement are briefly mentioned but should benefit from further elaboration and details, especially regarding how these critics handle complex test cases, also the correctness of CoT of LLM when evaluating the code patch.

2. **Limited Discussion on Scalability Constraints**: While the authors demonstrate strong results on SWE-bench, the generalizability of the approach to a broader set of software repositories and programming languages is not deeply explored. This leaves open questions regarding the framework’s ability to adapt to repositories that may require unique dependencies or multilingual support. Discussions on granularity (explanatory power), code size (upper limit of capability), and multi-domain (characterizing domain performance of LLM) of LLM in reviewing code patches are crucial but missing in this paper.

3. **Reliance on Gold Patches**: The framework's dependence on a reference (gold patch) for optimal accuracy raises potential issues in scenarios where a ground-truth patch may not exist. Although the authors attempt to address this with reference-free baselines, the performance drop observed here indicates that further research may be needed to refine reference-free evaluation methods.

**Questions:**

1. **Expand Baseline Comparisons**: The evaluation would be strengthened by including a broader set of baselines, potentially from benchmarks outside of SWE-bench. This could provide additional context for how well the framework generalizes across varied code types and patching workflows.

2. **Provide Example Outputs**: Including specific examples of code patches, along with LLM critics' predictions, would illustrate the system’s inner workings more vividly and clarify the practical implications of each evaluation level (e.g., context-enhanced vs. source code-based evaluations).

3. **In-depth Scalability Discussion**: Expanding the discussion on scalability, particularly in handling repositories with extensive interdependencies or complex testing setups, would provide insight into the framework’s applicability in large-scale industrial settings.

---

### Official Review · Reviewer_g42S · 2024-11-04

**Soundness:** 3
**Presentation:** 2
**Contribution:** 2
**Rating:** 5
**Confidence:** 5

**Summary:**

This paper proposes an execution-free, test-aware, LLM-based metric for evaluating code edits. Essentially, the authors prompt an LLM (claude-3-opus) with a candidate patch and an individual test in the test suite, and then the LLM will predict whether or not that patch will pass the given test. Then, they aggregate predictions across all tests in the test suit to assign a final build label. For experiments, the authors rely on SWE-Bench Lite, and trajectories form factory-code-droid, sweagent-gpt4, Gru, and code-story-aide-mixed. At the macro-level, their best approach achieves 71.4% accuracy (72.1 precision, 95.4 recall) with respect to predicting the build status outcome.

**Strengths:**

- The motivation for this work is quite nice and very important. An execution-free metric is definitely useful for fast iteration and in scenarios in which a test environment is not available.
- The finding that aggregating individual test results works better than having a holistic evaluation across all tests is quite interesting. This could possibly extend to other types of LLM-based evaluations as well (e.g., rather than evaluating across multiple dimensions in a single call, evaluate across each dimension independently and then aggregate results).
- The analysis with the model's self-reported confidence and test complexity is interesting.

**Weaknesses:**

- I believe an execution-free metric is most useful in situations in which you do not have a test suite at all or when the existing test suite has low coverage. However, this work requires having a high-quality test suite. The only dataset that the authors evaluate on is SWE-Bench, which comes with Docker images corresponding to the test environments, and so it seems like it is rather straightforward to just execute the tests in the test suite. Therefore, it seems that the impact of this work will be fairly limited.
- Additionally, from Table 2, the best accuracy that is attained is 71.4% (which is incorrectly claimed as 82.1% in the abstract). From the paper alone, I am not convinced that we can simply replace the execution-based metric with this. Perhaps if the authors had demonstrated that the rankings of the top ~10 models on the SWE-Bench leaderboard remained identical when using the LLM-based metric, it would have been more convincing. Currently, the best approach nearly matches the random baseline in terms of precision (i.e., the LLM-based approach will often say the patch passes tests when it actually does not). And if it is possible to obtain the execution-based score, then this LLM-based metric will likely not be needed at all since it serves to approximate the execution-based metric.
- A lot of details seem to be missing, misleading, inconsistent, and sometimes incorrect. This makes it difficult to follow and at times even assess the paper:

1. In the abstract, the authors claim an accuracy of 91.6% at the micro-level and 82.1% at the macro-level. However, these are F1 scores, and not accuracy, based on Tables 1 and 2.
2. For the majority of the first half of the paper, it seems that the authors use the gold code patch which resolves the issue (L017-018, L080-081, L137-141). However, it becomes clear later that the authors only consider the gold *test* patch and the gold code patch is not used in their approach and only in their baselines.
3. The prompts that the authors use are not given. Additionally, not a single example is provided.
4. The notion of "function-level" is in multiple tables but this is never explained. This also seems to be the best performing, so it is not clear what the best-performing method is actually. Namely, Table 1 has only two rows for "Isolated, Test-Aware", one corresponding to "post-commit functions" and one augmenting that method with "function-level." However, the two methods that are introduced in the main text are "context enhancement" and "source code" (which is also referred to as post-commit functions). In L259, it says "We can see that the LLM critic with context-enhanced patches performs the best, outperforming other baselines by 7.4% and 12.7%". But "context-enhanced" is not in Table 1 and there is only 1 baseline.
5. In the abstract, the authors motivate this work as an "intermediate/step-level" evaluation methodology. This suggests that the intermediate steps in the LLM-based agentic framework could be evaluated. However, this does not seem to be demonstrated in the paper. It is not obvious, but if Figure 4 (right) was intended to demonstrate this, it is not clear. Additionally, it is not clear why this was presented as a plot rather than an aggregate spearman's ranking coefficient. It looks like there is a decent chunk of the density in the negative range here too.

**Questions:**

- Please consider addressing the points raised above.
- Have you considered using both the gold test patch and gold code patch together?
- Did you consider a baseline which just uses the candidate code patch, with no tests at all?

---

### Meta-Review · Area_Chair_jRkS · 2024-12-23

**Metareview:**

This paper proposes an LLM-based “execution-free” metric for evaluating code edits. Rather than relying on an execution environment to run a test suite, the approach predicts per-test pass/fail outcomes using LLMs and then aggregates these predictions to derive an overall build status. Experiments are conducted on a subset of SWE-Bench (SWE-Bench Lite), comparing this LLM-based evaluation to execution-based and edit-distance baselines. The reported best macro-level performance is around 71% accuracy, suggesting partial viability of the approach as a proxy for execution-based evaluation.

**Strengths:**

* The motivation of using LLMs to predict test outcome on model-generated patches is “quite nice and very important” (g42S). This is an “under-explored problem” (3RwT),  and could be critical for scenarios where execution environments are not available (JASr, g42S).

* Interesting analysis on the results that aggregating individual test results work better than holistic evaluation (g42S). Similarly, 3RwT noted that “the micro-evaluation aggregation-based method is more sensible”.

* There are some interesting analysis in the paper, such as measuring self-reported confidence w.r.t. test complexity (g42S).

**Weaknesses:**

* **Questionable Practical Impact (g42S, EhhE)** The method requires having high-quality test suites, and the datasets used for experiments (SWE-Bench) already come with docker environments for test execution. It is questionable whether the proposed approach could actually generalize to more practical scenarios without tests or the tests are in low coverage, and execution environments are not readily available.

* There are several concerns regarding the accuracy of the LLM-based test outcome prediction approach. It is unclear whether this approach could replace execution-based metrics given that the best accuracy attained is around 71% (g42S). 3RwT similarly warns about overfitting, given that “SWE-Bench-Lite is imbalanced, with the majority of tests passing.”

* Multiple reviewers (g42S, JASr, EhhE) mention that certain parts of the method description are “misleading, inconsistent, and sometimes incorrect” (g42S).

Overall, while the idea of an execution-free metric is intriguing, the reviewers express significant concerns about the paper’s limited scope, unclear methodology, and questionable real-world utility. Decision: Reject.

**Additional Comments On Reviewer Discussion:**

The authors did not provide a response to the reviews.

---

### Decision · Program_Chairs · 2025-01-22

Reject